# Quantum circuit simulation of linear optics using fermion to qubit encoding

**Seungbeom Chin[1,2i⋆], Jaehee Kim[3i] and Joonsuk Huh[3,4,5†]**

**1** International Centre for Theory of Quantum Technologies (ICTQT),
University of Gdánsk, 80-308, Gdánsk, Poland
**2** Department of Electrical and Computer Engineering,
Sungkyunkwan University, Suwon 16419, Korea
**3** SKKU Advanced Institute of Nanotechnology (SAINT),
Sungkyunkwan University, Suwon 16419, Korea
**4** Department of Chemistry, Sungkyunkwan University, Suwon 16419, Korea
**5** Institute of Quantum Biophysics, Sungkyunkwan University, Suwon 16419, Korea

⋆ sbthesy@gmail.com , † joonsukhuh@gmail.com

## Abstract

This work proposes a digital quantum simulation protocol for the linear scattering process of bosons, which provides a simple extension to partially distinguishable boson cases. Our protocol is achieved by combining the boson-fermion correspondence relation and fermion to qubit encoding protocols. As a proof of concept, we designed quantum circuits for generating the Hong-Ou-Mandel dip by varying particle distinguishability. The circuits were verified with the classical and quantum simulations using the IBM Quantum and IonQ cloud services.

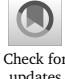

## Contents

[i]These authors contributed equally to the development of this work.

# 1 Introduction

Quantum simulation imitates an evolution of one quantum system with another artificially organized quantum system, i.e., quantum simulator [1]. Digital quantum simulators with qubits can encode an arbitrary quantum system comprising various particles, such as spins, fermions, and bosons, either exactly or approximately, depending on the particle nature. Qubits can be realized with several physical systems, such as trapped ions [2,3], nuclear magnetic resonance (NMR) [4,5], superconducting circuits [6,7], quantum dots [8], and photons [9]. Therefore, we can simulate any quantum system with digital quantum simulators using proper qubit encoding protocols regardless of the physical nature of the simulator.

Among various many-particle quantum systems, bosonic systems are considered to have the significant benefit from digital quantum simulations. Knill, Laflamme, and Milburn (KLM) showed that the postselected linear optics is capable of universal quantum computing [10]. Also, boson sampling proposed by Aaronson and Arkhipov [11] is a strong candidate for demonstrating the computational superiority of quantum devices. The boson sampling problem is believed to belong to classically hard sampling problems.

Inspired by the computational power of noninteracting bosonic systems, several boson to qubit encoding (B2QE) protocols have been proposed to simulate bosonic problems with digital quantum computers [12–18]. The majority of studies discretize bosonic creation and annihilation operators directly using unary or binary qubit representations of the Fock states as qubit encoding protocols. Ref. [15] presents a method for the digital quantum simulation of linear and nonlinear optical elements. Ref. [17] simulated the beam-splitting and squeezing operators with IBM Quantum based on the boson-qubit mapping developed in Ref. [19]. The required resources, such as the numbers of qubits and gates, vary according to the encoding protocols. Ref. [18] compared the resource efficiency among encoding protocols.

In this paper, we propose an alternative many-boson digital simulation method by combining the boson-fermion correspondence analyzed by Shchesnovich [20] and fermion to qubit encoding (F2QE) protocols [21, 22]. Specifically, *our protocol transforms bosonic states into fermionic states with internal degrees of freedom, which are then transformed to qubit states via a F2QE protocol (Jordan-Wigner (JW) transformation)*. With our simulation model, quantum circuits with $M$ bundles of $N$ qubits can simulate the number-conserving scattering process of $N$ bosons in $M$ modes. Our protocol is summarized in Fig. 1. The most significant advantage of our protocol is that *it can efficiently simulate non-ideal partially distinguishable bosons, i.e., bosons with internal degrees of freedom, using a direct extension of qubit numbers.*

As a proof of concept, we generate the Hong-Ou-Mandel (HOM) dip [23] with our protocol. The HOM effect is important in optical quantum systems that provide the elementary resource for logic gates in the linear optical quantum computing systems. The formal connection between the HOM effect and the qubit-based SWAP test was discussed in Ref. [24]. To simulate HOM dip, we need a method to add an internal degree of freedom to photons. It is easily achieved in our case by increasing the qubit number twice, which shows that our protocol is suitable for simulating partially distinguishable bosons. We verified the validity of our circuit using the IBM Quantum and IonQ cloud services.

This paper is organized as follows: Section 2 explains our digital boson simulation protocol. After reviewing the boson-fermion transformation protocol, we show how to combine this transformation with the JW transformation for the digital bosonic simulation. In section 3, we apply our model to the HOM dip experiment. We simulate the two-photon partial distinguishability with an eight-qubit-circuit. Finally, section 4 concludes our present work and discusses its possible future extensions.

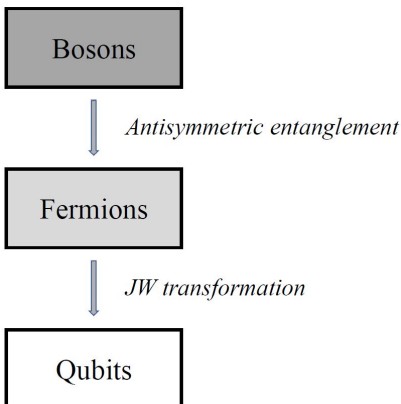

Figure 1: Our protocol for digital simulation of multi-boson systems. Using antisymmetrically entangled fermions as effective bosons, we can design digital quantum circuit to simulate multi-bosonic system via JW transformation.

## 2 Digitizing bosonic systems

In this section, we explain our B2QE protocol to simulate many-boson systems with qubits. Our protocol consists of two steps: First, we express the number-conserving bosonic systems with entangled multi-fermions with an internal degree of freedom. Second, we map the translated multi-fermionic system to a qubit system using a well-known F2QE protocol, the JW transformation [21].

### 2.1 Effective bosonic states of multi-fermions

We first explain how a specific form of entangled multi-fermions can effectively behave as multi-bosons. In the second quantization language, the bosonic creation and annihilation operators $\hat{a}_i^\dagger$ and $\hat{a}_i$ ($i = 1, \cdots, M$) obey the following commutation relations:

$$[\hat{a}_i, \hat{a}_j^\dagger] = \delta_{ij}, \qquad [\hat{a}_i, \hat{a}_j] = [\hat{a}_i^\dagger, \hat{a}_j^\dagger] = 0, \tag{1}$$

while the fermionic operators $\hat{b}_i^\dagger$ and $\hat{b}_i$ obey the anti-commutation relations:

$$\left\{\hat{b}_i, \hat{b}_j^\dagger\right\} = \delta_{ij}, \qquad \left\{\hat{b}_i, \hat{b}_j\right\} = \left\{\hat{b}_i^\dagger, \hat{b}_j^\dagger\right\} = 0, \tag{2}$$

where $\{\hat{A}, \hat{B}\} \equiv \hat{A}\hat{B} + \hat{B}\hat{A}$. The above relations satisfy the Pauli exclusion principle for fermions, which prohibits the superposition of two fermions in the same state. Indeed, we see that $\hat{b}_i^\dagger \hat{b}_i^\dagger |vac\rangle = -\hat{b}_i^\dagger \hat{b}_i^\dagger |vac\rangle = 0$ by Eq. (2), where $|vac\rangle$ is a vacuum state. On the other hand, if the fermions have internal degrees of freedom, such as spin, fermionic modes with different internal states can occupy the same spatial mode. By denoting a $K$-dimensional internal degree of freedom as $\mu$ ($\mu = 0, \cdots K - 1$), a fermionic operator with internal degrees of freedom $\mu$ is defined as $\hat{b}_i^{\dagger\mu}$ and $\hat{b}_i^\mu$. The anticommutation relations for the operators are as follows:

$$\left\{\hat{b}_i^\mu, \hat{b}_j^{\dagger\nu}\right\} = \delta_{ij}\delta^{\mu\nu}, \qquad \left\{\hat{b}_i^\mu, \hat{b}_j^\nu\right\} = \left\{\hat{b}_i^{\dagger\mu}, \hat{b}_j^{\dagger\nu}\right\} = 0. \tag{3}$$

In such a case, the fermions can condensate in the same spatial mode up to $K$. We aim to employ this feature of multi-fermionic states for mimicking the Bose-Einstein condensation (BEC) with the cutoff $K$. Fig. 2 explains the concept of fermionic condensation.

On the other hand, for the fermionic condensation to operate like the BEC, we must properly consider the fundamental differences between bosons and fermions, i.e., the exchange

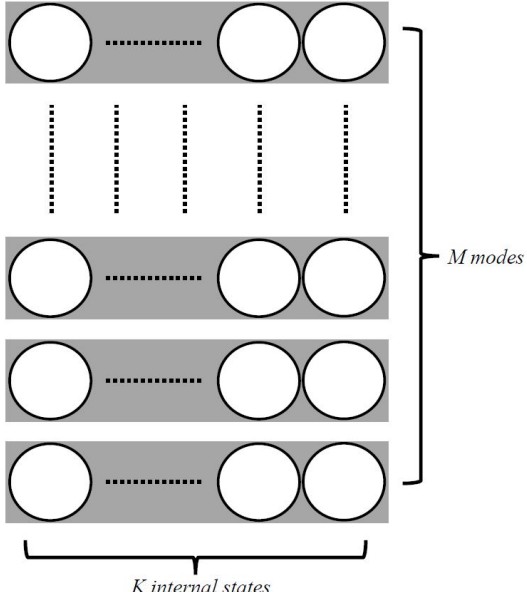

Figure 2: $M$-fermionic modes with $K$ internal states. Fermions can condensate in the same mode up to $K$. If a fermion state is entangled in Eq. (5), we can simulate BEC with the condensation cutoff $K$.

symmetry and antisymmetry indicated in Eqs. (1) and (2). Shchesnovich [20] showed that the interchangeability of entanglement and exchange symmetry can render entangled multi-fermions symmetric under the exchange of spatial modes. Here, we introduce *the effective bosonic state of multi-fermions with the condensation limit K* in the second quantization language, which offers a more refined explanation than of the first quantization language used in Ref. [20].

Let us consider an $N$-fermionic state,

$$\hat{b}_{i_1}^{\dagger \mu_1}\hat{b}_{i_2}^{\dagger \mu_2}\cdots \hat{b}_{i_N}^{\dagger \mu_N}|vac\rangle \tag{4}$$

($i_\alpha = 1, 2, \cdots, M$ and $\mu_\alpha = 1, 2, \cdots, K$ for $1 \leq \alpha \leq N$). This state is always antisymmetric under the exchange of the total indices $(\mu, i)$. However, if $K \geq N$, we can obtain a symmetric state under the spatial modes $i_\alpha$ by suitably superposing fermionic states as follows:

$$\frac{1}{\sqrt{N!}}\hat{b}_{i_1}^{\dagger [\mu_1}\hat{b}_{i_2}^{\dagger \mu_2}\cdots \hat{b}_{i_N}^{\dagger \mu_N]}|vac\rangle \tag{5}$$

(a square bracket [ , ] on the upper indices means that the indices are antisymmetrized. For the simplest example, $\hat{b}_{i_1}^{\dagger [\mu_1}\hat{b}_{i_2}^{\dagger \mu_2]} \equiv \hat{b}_{i_1}^{\dagger \mu_1}\hat{b}_{i_2}^{\dagger \mu_2} - \hat{b}_{i_1}^{\dagger \mu_2}\hat{b}_{i_2}^{\dagger \mu_1}$). Since the following relation,

$$\hat{b}_{i_1}^{\dagger [\mu_1}\cdots \hat{b}_{i_\alpha}^{\dagger \mu_\alpha}\cdots \hat{b}_{i_\beta}^{\dagger \mu_\beta}\cdots \hat{b}_{i_N}^{\dagger \mu_N]}|vac\rangle = -\hat{b}_{i_1}^{\dagger [\mu_1}\cdots \hat{b}_{i_\beta}^{\dagger \mu_\beta}\cdots \hat{b}_{i_\alpha}^{\dagger \mu_\alpha}\cdots \hat{b}_{i_N}^{\dagger \mu_N]}|vac\rangle$$
$$= \hat{b}_{i_1}^{\dagger [\mu_1}\cdots \hat{b}_{i_\beta}^{\dagger \mu_\alpha}\cdots \hat{b}_{i_\alpha}^{\dagger \mu_\beta}\cdots \hat{b}_{i_N}^{\dagger \mu_N]}|vac\rangle, \tag{6}$$

holds for any $\alpha$ and $\beta$ for $1 \leq \alpha \leq N$ and $1 \leq \beta \leq N$, we have

$$\frac{1}{\sqrt{N!}}\hat{b}_{i_1}^{\dagger [\mu_1}\hat{b}_{i_2}^{\dagger \mu_2}\cdots \hat{b}_{i_N}^{\dagger \mu_N]}|vac\rangle = \frac{1}{\sqrt{N!}}\hat{b}_{\{i_1}^{\dagger [\mu_1}\hat{b}_{i_2}^{\dagger \mu_2}\cdots \hat{b}_{i_N\}}^{\dagger \mu_N]}|vac\rangle \tag{7}$$

(a brace { , } on the lower indices on the right hand side denotes that the indices are symmetrized. For the simplest example, $\hat{b}_{\{i_1}^{\dagger \mu_1}\hat{b}_{i_2\}}^{\dagger \mu_2} \equiv \hat{b}_{i_1}^{\dagger \mu_1}\hat{b}_{i_2}^{\dagger \mu_2} + \hat{b}_{i_2}^{\dagger \mu_1}\hat{b}_{i_1}^{\dagger \mu_2}$). Therefore, we can consider Eq (5) to be an effective $N$-boson state with the condensation limit $K$.

As a simple example, when $N = 2$, Eq. (5) becomes

$$\frac{1}{\sqrt{2}}\left(\hat{b}_{i_1}^{\dagger\mu_1}\hat{b}_{i_2}^{\dagger\mu_2} - \hat{b}_{i_1}^{\dagger\mu_2}\hat{b}_{i_2}^{\dagger\mu_1}\right)|vac\rangle. \tag{8}$$

By exchanging the mode indices $i_1$ and $i_2$, we have

$$\frac{1}{\sqrt{2}}\left(\hat{b}_{i_2}^{\dagger\mu_1}\hat{b}_{i_1}^{\dagger\mu_2} - \hat{b}_{i_2}^{\dagger\mu_2}\hat{b}_{i_1}^{\dagger\mu_1}\right)|vac\rangle = \frac{1}{\sqrt{2}}\left(-\hat{b}_{i_1}^{\dagger\mu_2}\hat{b}_{i_2}^{\dagger\mu_1} + \hat{b}_{i_1}^{\dagger\mu_1}\hat{b}_{i_2}^{\dagger\mu_2}\right)|vac\rangle$$

$$= \frac{1}{\sqrt{2}}\left(\hat{b}_{i_1}^{\dagger\mu_1}\hat{b}_{i_2}^{\dagger\mu_2} - \hat{b}_{i_1}^{\dagger\mu_2}\hat{b}_{i_2}^{\dagger\mu_1}\right)|vac\rangle, \tag{9}$$

where the second line is obtained by changing the order of fermionic operators.

Since the antisymmetrical entanglement of the fermions is essential for effective bosonic states to behave like bosons, the exchange symmetry of the state must be preserved under evolutions. In other words, if we want to simulate the bosonic scattering process with fermions, the transformation operators of fermions must preserve the antisymmetrical entanglement. We observe that some transformation operators satisfy this restriction. We first consider a bosonic operator $T$ of the following form:

$$T = \exp\left[it\left(\sum_{j,k=1}^{M}\Phi_{jk}\hat{a}_j^{\dagger}\hat{a}_k\right)\right], \tag{10}$$

where $t$ is the evolution time and $\Phi_{jk} \in \mathbb{C}$. We note that

$$\sum_{jk}\Phi_{jk}\hat{a}_j^{\dagger}\hat{a}_k \tag{11}$$

behaves as the Hamiltonian of the given system by setting $\Phi_{jk} = \Phi_{kj}^*$. Then, the transformation of $\hat{a}_i^{\dagger}$ under $T$ is given by

$$T\hat{a}_i^{\dagger}T^{\dagger} = \sum_j \exp(it\Phi^*)_{ij}\hat{a}_j^{\dagger}$$

$$\equiv \sum_j u_{ij}\hat{a}_j^{\dagger}, \tag{12}$$

where $\Phi$ is a Hermitian matrix whose elements are $\Phi_{ij}$ and $\sum_j u_{ij}u_{kj}^* = \delta_{ik}$. In the fermionic system, the corresponding operator $T_f$ is expressed as follows:

$$T_f = \exp\left(it\sum_{\mu}\sum_{j,k}\Phi_{jk}\hat{b}_j^{\dagger\mu}\hat{b}_k^{\mu}\right), \tag{13}$$

which gives

$$T_f\hat{b}_i^{\dagger\mu}T_f^{\dagger} = \sum_j u_{ij}\hat{b}_j^{\dagger\mu}. \tag{14}$$

Then, the state Eq. (5) evolves via $T_f$ as follows:

$$|\Psi\rangle_f = \frac{1}{\sqrt{N!}}\sum_{k_1,\cdots,k_N} u_{\{j_1}^{k_1}\cdots u_{j_N\}}^{k_N}\hat{b}_{k_1}^{\dagger[\mu_1}\cdots\hat{b}_{k_N}^{\dagger\mu_N]}|vac\rangle$$

$$= \frac{1}{\sqrt{N!}}\sum_{k_1,\cdots,k_N} u_{\{j_1}^{\{k_1}\cdots u_{j_N\}}^{k_N\}}\hat{b}_{\{k_1}^{\dagger[\mu_1}\cdots\hat{b}_{k_N\}}^{\dagger\mu_N]}|vac\rangle. \tag{15}$$

The second line of the above equation shows that the transformed state is a linear combination of effective multi-boson states, which itself is an effective multi-boson state. In a more general form, we see that any number-conserving Hamiltonian looks like $H = \sum_{jk} \Phi_{jk} \hat{a}_j^\dagger \hat{a}_k + \text{c.c.}$.

Finally, we check whether the measurement of the state Eq. (5) that evolves with Eq. (10) is effectively bosonic, i.e., the scattering probability is proportional to the absolute square of the transformation matrix permanent. Suppose first that we postselect terms without bunching, irrespective of what the internal states of the particles are. Without loss of generality, we can assume the boson number distribution vector as follows:

$$\vec{n} = (\underbrace{1, 1, \cdots, 1}_{N}, \underbrace{0, \cdots, 0}_{M-N}) \qquad (M \geq N). \tag{16}$$

Then, the scattering probability is given with a projector

$$E = \sum_{\mu_1 \cdots \mu_N} (\hat{b}_1^{\dagger \mu_1}) \cdots \hat{b}_N^{\dagger \mu_N} |vac\rangle \langle vac| \hat{b}_N^{\mu_N} \cdots \hat{b}_1^{\mu_1},$$

as follows:

$$\begin{aligned}
P &= \text{Tr}(E \rho_f) \\
&= \sum_{\mu_1, \cdots, \mu_N} \langle vac| \hat{b}_N^{\mu_N} \cdots \hat{b}_1^{\mu_1} |\Psi\rangle \langle \Psi|_f \hat{b}_1^{\dagger \mu_1} \cdots \hat{b}_N^{\dagger \mu_N} |vac\rangle.
\end{aligned} \tag{17}$$

Using the relation,

$$\hat{b}_{i_1}^{\mu_1} \cdots \hat{b}_{i_N}^{\mu_N} \hat{b}_{k_1}^{\dagger [\nu_1} \cdots \hat{b}_{k_N}^{\dagger \nu_N]} |vac\rangle = \delta_{i_1}^{\{k_1} \cdots \delta_{i_N}^{k_N\}} \delta_{\mu_1}^{[\nu_1} \cdots \delta_{\mu_N}^{\nu_N]} |vac\rangle, \tag{18}$$

we have

$$P \sim |\text{perm}(u)|^2, \tag{19}$$

where $u$ is an $N \times N$ matrix whose entries are $u_{ij}$ and $\text{perm}(u)$ denotes the permanent of $u$, as expected for a bosonic systems with $T$ [11,25]. If the postselected states permit bunching, the probability becomes proportional to the permanent of the submatrix of $u$ as expected [26–29].

## 2.2 Simulating multi-boson systems with qubits

Since a fermionic state of the form indicated in Eq. (5) can simulate a linear scattering of bosons, we conclude that digital quantum computers can also simulate the same system using the JW transformation. Before explaining how we actually organize quantum circuits and algorithms for such a simulation, we first review the JW transformation, which maps fermions to qubits [21].

In the JW transformation, qubit states $|0\rangle$ and $|1\rangle$ correspond to the empty and occupied states of fermions for a given mode, i.e., the following isomorphism should hold:

$$\begin{aligned}
N \text{ qubit state } |\vec{n}\rangle &= |n_1, \cdots, n_N\rangle \qquad (n_j = 0, 1) \\
&\cong \quad N \text{ fermionic state } (\hat{b}_1^\dagger)^{n_1} \cdots (\hat{b}_1^\dagger)^{n_N} |vac\rangle.
\end{aligned} \tag{20}$$

The left and right hand side denotes an $N$-qubit state and an $N$-fermionic state, respectively, and $\cong$ represents that the two sides are in a correspondence relationship with each other. For this relationship to hold, there must be operators acting on the $N$-qubit system that play the roles of creation and annihilation operators. Indeed, we can construct such operators by combining the Pauli operators $X_j, Y_j$ and $Z_j$ ($j = 1, \cdots L$), i.e., $\hat{b}_j^\dagger(X, Y, Z) \cong \hat{b}_j^\dagger$ and $\hat{b}_j(X, Y, Z) \cong \hat{b}_j$.

We can see that $|\vec{n}\rangle$ and $b_j^\dagger(X, Y, Z)$ must satisfy the following conditions:

- If $n_j = 0$, then $\hat{b}_j|\vec{n}\rangle = 0$

- If $n_j = 1$, then $\hat{b}_j|\vec{n}\rangle = (-1)^{s_{\vec{n}}^j + 1}|n_1, \cdots, n_j \oplus 1, \cdots, n_L\rangle$ where $s_{\vec{n}}^j \equiv \sum_{k=1}^{j-1} n_k$. Note that $(-1)^{s_{\vec{n}}^j + 1}$ comes from the anticommutation property of the creation-annihilation operators.

It can easily be verified that

$$
\begin{aligned}
\hat{b}_j(X, Y, Z) &\equiv (\otimes_{k=1}^{j-1} Z_k) \otimes \sigma_j^-, \\
\hat{b}_j^\dagger(X, Y, Z) &\equiv (\otimes_{k=1}^{j-1} Z_k) \otimes \sigma_j^+
\end{aligned}
\tag{21}
$$

($\sigma^+ \equiv |1\rangle\langle 0|$ and $\sigma^- \equiv |0\rangle\langle 1|$), satisfy the above conditions. One can also check that Eq. (21) satisfies the anticommutation relations, i.e., $\{\hat{b}_j, \hat{b}_k^\dagger\} = \delta_{jk}$ and $\{\hat{b}_j^\dagger, \hat{b}_k^\dagger\} = \{\hat{b}_j, \hat{b}_k\} = 0$. The state transformation of Eq. (20) and operator transformations in Eq. (21) define the JW transformation for the digital simulation of fermionic systems.

By combining the JW transformation and the results of Section 2.1, we can see that $N$ bosons in $M$ modes can be simulated with $NM$ qubits (see Fig. 3). To impose this correspondence, consider an $MN$-qubit state

$$
\left|\left(n_1^1, \cdots, n_1^N\right), \left(n_2^1, \cdots, n_2^N\right), \cdots, \left(n_M^1, \cdots, n_M^N\right)\right\rangle,
\tag{22}
$$

where $n_i^\mu = 0, 1$ and each bracket $(n_i^1, n_i^2, \cdots, n_i^N)$ denotes the state of a bundle of $N$ qubits.

If $n_i^{\mu'} = 1$, then it is considered in the fermion picture that a fermion exists in the $i$th mode with internal state $j$. Any state of this kind can be generated from $|vac\rangle \cong |\underbrace{00\cdots0}_{N \times M}\rangle$ with the creation operators as follows:

$$
\begin{aligned}
\hat{b}_1^{\dagger 1} &= \sigma^+, \\
\hat{b}_1^{\dagger 2} &= Z \otimes \sigma^+, \\
&\vdots \\
\hat{b}_1^{\dagger N} &= \underbrace{Z \otimes \cdots \otimes Z}_{N-1} \otimes \sigma^+, \\
&\vdots \\
\hat{b}_M^{\dagger N} &= \underbrace{Z \otimes \cdots \otimes Z}_{NM-1} \otimes \sigma^+.
\end{aligned}
\tag{23}
$$

Now we can express an effective multi-boson state described in Eq. (5), which is entangled as antisymmetric under the internal states in the qubit space. As an example, consider the case with $N$ bosons when all bosons from 1 to $N$ are in different modes with respect to each other. Using Eq. (7), such a state can be expressed as $\frac{1}{\sqrt{N!}}\hat{b}_1^{\dagger[1}\cdots\hat{b}_N^{\dagger N]}|vac\rangle$. By defining $\chi_i = (\underbrace{0, \cdots, 0, \overset{i\text{th}}{1}, 0, \cdots, 0}_{N})$ and $\chi_0 = (\underbrace{0, 0, \cdots, 0}_{N})$, the state can be expressed in the $NM$ qubit space as follows:

$$
\frac{1}{\sqrt{N!}}\sum_{\rho \in S_N} sgn(\rho)|\chi_{\rho(1)}, \chi_{\rho(2)}, \cdots, \chi_{\rho(N)}, \chi_0, \cdots, \chi_0\rangle,
\tag{24}
$$

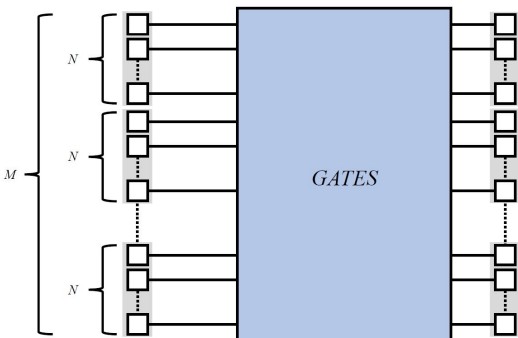

Figure 3: $NM$ qubits that can simulate $N$ bosons in $M$ modes. Each bundle of $N$ qubits behaves as a mode that can contain up to $N$ bosons. Using $M$ bundles of $N$ qubits, we can simulate $N$-boson scattering process in $M$ modes.

where $S_N$ is the permutation group. On the other hand, if all the bosons are in the same mode, e.g., the first mode, the state can be written as follows:

$$\frac{1}{\sqrt{N!}}\hat{b}_1^{\dagger[1]}\cdots\hat{b}_1^{\dagger[N]}|vac\rangle = \hat{b}_1^{\dagger 1}\cdots\hat{b}_1^{\dagger N}|vac\rangle$$
$$\cong |(\underbrace{1,1,\cdots,1}_{N}),\chi_0,\cdots,\chi_0\rangle. \tag{25}$$

For the case with $N = 2$ and $M = 3$, Eq. (24) takes the following form:

$$\frac{1}{\sqrt{2}}\big(|\chi_1,\chi_2,\chi_0\rangle - |\chi_2,\chi_1,\chi_0\rangle\big) = \frac{1}{\sqrt{2}}\big(|10,01,00\rangle - |01,10,00\rangle\big), \tag{26}$$

which corresponds to the bosonic state $\hat{a}_1^\dagger\hat{a}_2^\dagger|vac\rangle$, while Eq. (25) becomes $|11,00,00\rangle$, which corresponds to the bosonic state $\frac{1}{\sqrt{2}}(\hat{a}_1^\dagger)^2|vac\rangle$.

Since Eqs. (22) and (23) represent a mapping from bosonic systems to qubits, we can digitally simulate multi-boson systems with the following process:

1. Preparation of the initial state: We first need to prepare the initial states of the form shown in Eq. (7), which can be achieved by adopting one of the known antisymmtrization algorithms, e.g., those in Refs. [30,31]. On the other hand, we can find optimal algorithms for the states with small $N$ case-by-case.

2. Evolution: The unitary operations can be executed by substituting Eq. (23) into the Hamiltonian operator of Eq. (11).

3. Measurement: While the order of the excited states is unimportant, the number of excited states in each bundle is crucial because it determines the distributions of boson numbers. For example, if $N = 3$, (100), (010), and (001), in all cases a mode has one particle with different internal state. Nevertherless, we only record that one of three qubit states in the bundle is excited. Eq. (17) represents such a measurement process.

## 3 Application: Hong-Ou-Mandel dip

In this section, we use our protocol to simulate the HOM effect for $N = 2$ [23]. We first simulate ideal photon case (with no internal degree of freedom), which is then generalized to non-ideal photons with a two-dimensional internal degree of freedom. This generalization shows our protocol can simulate non-ideal bosons simply with a direct extension of qubit numbers.

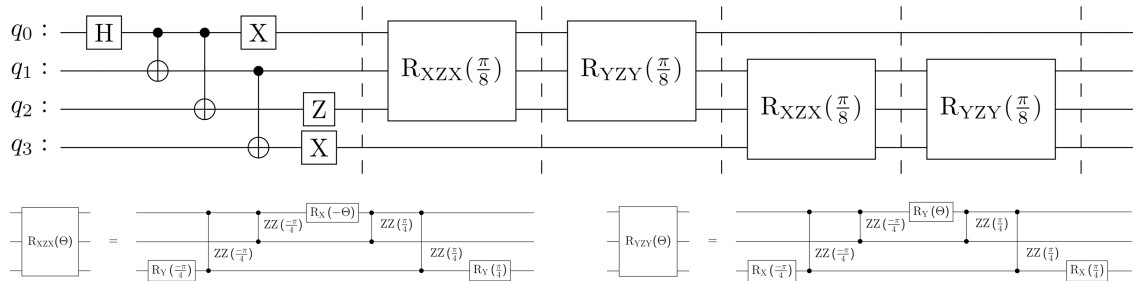

Figure 4: Full circuit for HOM experiment. As seen in figures in the second line, $T_H$ given by Eq. (32) is further decomposed into one- or two-qubit gates. We set, for example, $R_{XZX}(\Theta) = \exp[i\Theta(X \otimes Z \otimes X)]$, where the index indicates the operator in the exponent.

## 3.1 HOM experiment with ideal photons

Since two qubits can represent a bosonic mode with a maximal photon number of two, our protocol needs four qubits here.

**Preparation.—** Using the notations given before Eq. (24), we prepare the following initial state $|\Psi\rangle_i$:

$$|\Psi\rangle_i = \frac{1}{\sqrt{2}}\big(|\chi_1, \chi_2\rangle - |\chi_2, \chi_1\rangle\big)$$
$$= \frac{1}{\sqrt{2}}\big(|10, 01\rangle - |01, 10\rangle\big). \tag{27}$$

**Evolution.—** For the case of HOM scattering, we set

$$t = \frac{\pi}{4}, \qquad \Phi = \begin{pmatrix} 0 & 1 \\ 1 & 0 \end{pmatrix}, \tag{28}$$

in Eq. (10) which produces the following transformation operator $T_H$:

$$T^H \equiv \exp\left[\frac{i\pi}{4}\big(\hat{a}_1^\dagger \hat{a}_2 + \hat{a}_2^\dagger \hat{a}_1\big)\right]. \tag{29}$$

In the fermion system, $T_H$ is given as follows:

$$T_f^H = \exp\left[\frac{i\pi}{4}\big(\hat{b}_1^{\dagger 1}\hat{b}_2^1 + \hat{b}_2^{\dagger 1}\hat{b}_1^1 + \hat{b}_1^{\dagger 2}\hat{b}_2^2 + \hat{b}_2^{\dagger 2}\hat{b}_1^2\big)\right]$$
$$= \exp\left[\frac{i\pi}{4}\big(\hat{b}_1^{\dagger 1}\hat{b}_2^1 + \hat{b}_2^{\dagger 1}\hat{b}_1^1\big)\right]\exp\left[\frac{i\pi}{4}\big(\hat{b}_1^{\dagger 2}\hat{b}_2^2 + \hat{b}_2^{\dagger 2}\hat{b}_1^2\big)\right]. \tag{30}$$

Using the JW transformation, we obtain

$$\hat{b}_1^{\dagger 1}\hat{b}_2^1 + \hat{b}_2^{\dagger 1}\hat{b}_1^1 = \frac{1}{2}(X \otimes Z \otimes X + Y \otimes Z \otimes Y) \otimes \mathbb{I},$$
$$\hat{b}_1^{\dagger 2}\hat{b}_2^2 + \hat{b}_2^{\dagger 2}\hat{b}_1^2 = \frac{1}{2}\mathbb{I} \otimes (X \otimes Z \otimes X + Y \otimes Z \otimes Y), \tag{31}$$

in the qubit system. Since $X \otimes Z \otimes X$ and $Y \otimes Z \otimes Y$ commute, $T_H$ can be further decomposed follows:

$$T_H = \exp\left[\frac{i\pi}{8}(X \otimes Z \otimes X \otimes \mathbb{I})\right]\exp\left[\frac{i\pi}{8}(Y \otimes Z \otimes Y \otimes \mathbb{I})\right]$$
$$\times \exp\left[\frac{i\pi}{8}(\mathbb{I} \otimes X \otimes Z \otimes X)\right]\exp\left[\frac{i\pi}{8}(\mathbb{I} \otimes Y \otimes Z \otimes Y)\right]. \tag{32}$$

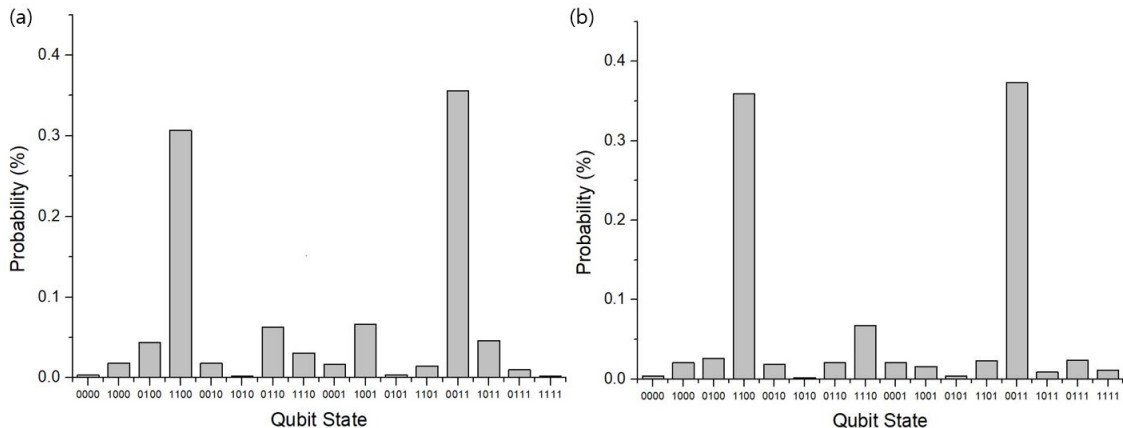

Figure 5: Simulation results of the circuit in Fig. 4 using quantum devices: (a) ibm_brisbane of IBM Quantum, and (b) ionq_qpu of IONQ.

Note that we have not used the Trotter decomposition, because all the terms in the exponential terms commute with each other. This is true for the general linear optical transformations [15].

**Measurement.—** The final state transformed by Eq. (29) is given by

$$|\Psi\rangle_f = \frac{i}{\sqrt{2}}\big(|11,00\rangle + |00,11\rangle\big). \tag{33}$$

The interpretation of the above state is that two bosons always bunch, i.e., the HOM effect occurs.

The full circuit for the HOM digital simulation is shown in Fig. 4. According to Eq. (33), the ideal outcome probabilities of states $|11,00\rangle$ and $|00,11\rangle$ are 50% each, which shows the photon-bunching effect of indistinguishable photons. The ideal result can be confirmed via classical simulations of Fig. 4 using, for example, Qiskit's qasm_simulator. We used ibm_brisbane of IBM Quantum and ionq_qpu of IONQ for the digital quantum simulation of the ideal indistinguishable photons case. Fig. 5 shows the results. In contrast to the theoretical prediction, quantum states other than $|11,00\rangle$ and $|00,11\rangle$ were measured. It is attributed to errors arising from gate operations and measurements in real quantum devices. For assessment of the simulation, we performed fidelity calculations after tomography for ibm_brisbane, ibm_perth, ibm_lagos, and ibm_nairobi on IBM Quantum, and ionq_qpu on IONQ. We summarized the tomography and fidelity calculation results in Supplementary Materials, along with the technical specifications of the devices used.

## 3.2 HOM dip

We will now simulate the HOM dip (see, e.g., [32] for a pedagogic review) with a two-dimensional internal degree of freedom that creates distinguishability. By denoting the internal state of bosons as $s$ $(= 0, 1)$, the creation and annihilation operators are written as $\hat{a}^\dagger_{is}$ and $\hat{a}_{is}$ with $[\hat{a}_{is}, \hat{a}^\dagger_{jr}] = \delta_{ij}\delta_{sr}$. Then, an $N$-boson state $\hat{a}^\dagger_{i_1 s_1} \hat{a}^\dagger_{i_2 s_2} \cdots \hat{a}^\dagger_{i_N s_N}|vac\rangle$ ($i_\alpha \in \{1, \cdots, N\}$, $s_\beta \in \{0, 1\}$ for $\alpha, \beta \in \{1, \cdots, N\}$) can effectively be expressed as a fermionic state as follows:

$$\frac{1}{\sqrt{N!}} \hat{b}^{\dagger[\mu_1}_{i_1 s_1} \hat{b}^{\dagger\mu_2}_{i_2 s_2} \cdots \hat{b}^{\dagger\mu_N]}_{i_N s_N}|vac\rangle. \tag{34}$$

Therefore, the general initial state for the HOM dip with two photons can be written as

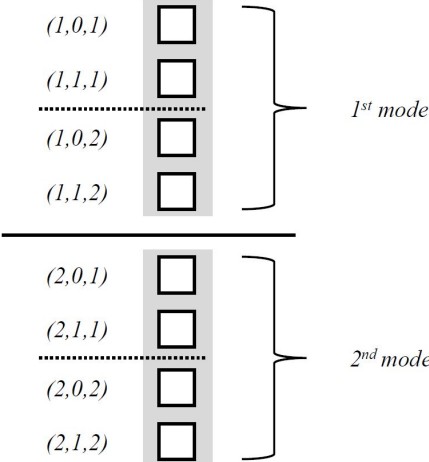

Figure 6: Qubit representation of two photons in two modes with two-dimensional internal degree of freedom.

follows:

$$|\Psi\rangle_i = \hat{a}^\dagger_{1|s\rangle}\hat{a}^\dagger_{2|r\rangle}|vac\rangle \cong \frac{1}{\sqrt{2}}\hat{b}^{\dagger[1]}_{1|s\rangle}\hat{b}^{\dagger[2]}_{2|r\rangle}|vac\rangle,\tag{35}$$

where $|s\rangle$ and $|r\rangle$ are the general internal states of the form $\zeta|0\rangle + \xi|1\rangle$ ($\zeta, \xi \in \mathbb{C}$ and $|\zeta|^2 + |\xi|^2 = 1$). To simulate this type of HOM dip, we need eight qubits, which are displayed in Fig. 6. Each qubit corresponds to the particle states $(i, \mu, s)$ as indicated in the figure.

**Preparation.—** Without loss of generality, we can assume the internal state of the photons as $|s\rangle = |0\rangle$ and $|r\rangle = \zeta|0\rangle + \xi|1\rangle$. Therefore, the initial state for partially distinguishable photons can be described as follows:

$$
\begin{aligned}
|\Psi\rangle_i &= \frac{1}{\sqrt{2}}\hat{b}^{\dagger[1]}_{10}\hat{b}^{\dagger[2]}_{2|r\rangle}|vac\rangle \\
&= \frac{1}{\sqrt{2}}\left(\zeta\hat{b}^{\dagger[1]}_{10}\hat{b}^{\dagger[2]}_{20} + \xi\hat{b}^{\dagger[1]}_{10}\hat{b}^{\dagger[2]}_{21}\right)|vac\rangle \\
&= \frac{1}{\sqrt{2}}\left(\zeta(|1000,0010\rangle - |0010,1000\rangle) + \xi(|1000,0001\rangle - |0010,0100\rangle)\right).
\end{aligned}\tag{36}
$$

We can prepare this state by first creating

$$\frac{1}{\sqrt{2}}\left(|1000,0010\rangle - |0010,1000\rangle\right),\tag{37}$$

and then applying the following gates:

$$
\begin{array}{c}
q_0: \quad \oplus \quad \bullet \quad \oplus \\
q_1: \quad \bullet \quad \boxed{U(\theta,\phi,\gamma)} \quad \bullet
\end{array}
$$

between $(2, 0, 1)$ and $(2, 1, 1)$ and between $(2, 0, 2)$ and $(2, 1, 2)$. The above gates can be represented in a matrix form as follows:

$$
\begin{pmatrix}
1 & 0 & 0 & 0 \\
0 & e^{i\gamma}\cos(\frac{\theta}{2}) & -e^{i\phi}\sin(\frac{\theta}{2}) & 0 \\
0 & e^{-i\phi}\sin(\frac{\theta}{2}) & e^{-i\gamma}\cos(\frac{\theta}{2}) & 0 \\
0 & 0 & 0 & 1
\end{pmatrix},\tag{38}
$$

where $\zeta$, and $\xi$ are given with $(\gamma, \phi, \theta)$ by

$$\zeta = e^{i\gamma} \cos\left(\frac{\theta}{2}\right), \qquad \xi = -e^{i\phi} \sin\left(\frac{\theta}{2}\right). \tag{39}$$

For two indistinguishable bosons (ideal photons), i.e., $\zeta = 1$, and the initial state is as follows:

$$\begin{aligned}
|\Psi\rangle_i^{ind} &= \frac{1}{\sqrt{2}} \hat{b}_{10}^{\dagger[1} \hat{b}_{20}^{\dagger2]} |vac\rangle \\
&= \frac{1}{\sqrt{2}} \left( |1000, 0010\rangle - |0010, 1000\rangle \right). \tag{40}
\end{aligned}$$

On the other hand, if two bosons are fully distinguishable, i.e., $\xi = 1$, the initial state can be given without loss of generality as follows:

$$\begin{aligned}
|\Psi\rangle_i^{dis} &= \frac{1}{\sqrt{2}} \hat{b}_{10}^{\dagger[1} \hat{b}_{21}^{\dagger2]} |vac\rangle \\
&= \frac{1}{\sqrt{2}} \left( |1000, 0001\rangle - |0010, 0100\rangle \right). \tag{41}
\end{aligned}$$

**Evolution.—** The evolution operator with distinguishability is simply obtained by generalizing Eq. (30) as follows:

$$\begin{aligned}
T_H &= \exp\left[ \frac{i\pi}{4} \sum_{s,\mu} \left( \hat{b}_{1s}^{\dagger\mu} \hat{b}_{2s}^{\mu} + \hat{b}_{2s}^{\dagger\mu} \hat{b}_{1s}^{\mu} \right) \right] \\
&= \exp\left[ \frac{i\pi}{4} \left( \hat{b}_{10}^{\dagger1} \hat{b}_{20}^{1} + \hat{b}_{20}^{\dagger1} \hat{b}_{10}^{1} \right) \right] \exp\left[ \frac{i\pi}{4} \left( \hat{b}_{11}^{\dagger1} \hat{b}_{21}^{1} + \hat{b}_{21}^{\dagger1} \hat{b}_{11}^{1} \right) \right] \\
&\quad \times \exp\left[ \frac{i\pi}{4} \left( \hat{b}_{10}^{\dagger2} \hat{b}_{20}^{2} + \hat{b}_{20}^{\dagger2} \hat{b}_{10}^{2} \right) \right] \exp\left[ \frac{i\pi}{4} \left( \hat{b}_{11}^{\dagger2} \hat{b}_{21}^{2} + \hat{b}_{21}^{\dagger2} \hat{b}_{11}^{2} \right) \right]. \tag{42}
\end{aligned}$$

**Measurement.—** When the bosons are indistinguishable, the final state $|\Psi\rangle_f^{ind}$ is as follows:

$$\begin{aligned}
|\Psi\rangle_f^{ind} &= \frac{i}{\sqrt{2}} \left( \hat{b}_{10}^{\dagger[1} \hat{b}_{10}^{\dagger2]} + \hat{b}_{20}^{\dagger[1} \hat{b}_{20}^{\dagger2]} \right) |vac\rangle \\
&= \frac{i}{\sqrt{2}} \left( |1010, 0000\rangle + |0000, 1010\rangle \right), \tag{43}
\end{aligned}$$

i.e., two particles are always in the same mode and the coincidence probability (the probability that each mode simultaneously detect particles) becomes zero.

When they are distinguishable, the final state $|\Psi\rangle_f^{dis}$ is given as follows:

$$\begin{aligned}
|\Psi\rangle_f^{dis} &= \frac{1}{\sqrt{2}} \left( i\hat{b}_{10}^{\dagger[1} \hat{b}_{11}^{\dagger2]} + \hat{b}_{10}^{\dagger[1} \hat{b}_{21}^{\dagger2]} - \hat{b}_{20}^{\dagger[1} \hat{b}_{11}^{\dagger2]} + i\hat{b}_{20}^{\dagger[1} \hat{b}_{21}^{\dagger2]} \right) |vac\rangle \\
&= \frac{i}{\sqrt{2}} \Big[ i\left( |1001, 0000\rangle - |0110, 0000\rangle \right) + i\left( |0000, 1001\rangle - |0000, 0110\rangle \right) \\
&\quad + \left( |1000, 0001\rangle - |0010, 0100\rangle \right) - \left( |0001, 1000\rangle - |0100, 0010\rangle \right) \Big], \tag{44}
\end{aligned}$$

which means that each particle can arrive at each of the two detectors with probability 0.5.

In general, the final state with an arbitrary distinguishability ($|r\rangle = \zeta|0\rangle + \xi|1\rangle$) is given by

$$\begin{aligned}
|\Psi\rangle_f &= \frac{i\zeta}{\sqrt{2}} \left( |1010, 0000\rangle + |0000, 1010\rangle \right) \\
&\quad + \frac{\xi}{2\sqrt{2}} \Big( i\left( |1001, 0000\rangle - |0110, 0000\rangle \right) + \left( |1000, 0001\rangle - |0010, 0100\rangle \right) \\
&\quad - \left( |0001, 1000\rangle - |0100, 0010\rangle \right) + i\left( |0000, 1001\rangle - |0000, 0110\rangle \right) \Big). \tag{45}
\end{aligned}$$

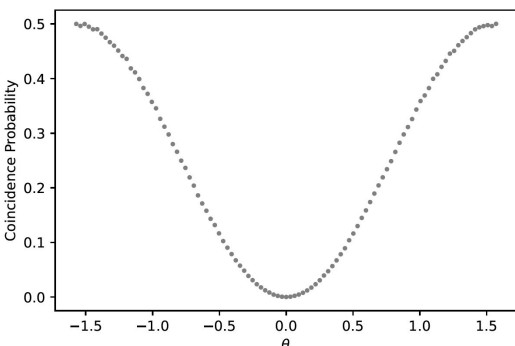

Figure 7: HOM dip classical simulation graph according to variation of $\theta$ from $-\pi$ to $\pi$. The interval between angles is $\pi/100$. All simulations were performed using Qiskit's qasm_simulator.

We can predict that the coincidence probability for the photons varies from 0 (fully indistinguishable) to 0.5 (fully distinguishable).

The increase in the circuit's width corresponds to the rise in the depth for simulating the scattering process of partially distinguishable photons, which causes a significant error in the quantum simulation. Therefore, we executed a classical simulation with qasm to show the validity of our method. Fig. 7 reveals a clear pattern of the HOM dip.

**Remark.—** It is worth mentioning that our example of the HOM dip simulation with distinguishable bosons shows the advantage of our scheme over other integer-to-bit mappings in, e.g., Ref. [13–19]. Comparing Eq. (27) with Eq. (36), we see that bosonic system with a 2-dimensional internal degree of freedom is directly simulated by adding one copy of 4 qubits. Since our mapping from the bosonic system to qubits is set to preserve the exchange symmetry, the generalization from ideal bosons to distinguishable bosons is straightforward. Moreover, we do not need Schur transformation gates as in Ref. [12], hence more efficient. On the other hand, recent research on the digital simulation of the HOM experiment with ideal photons in Ref. [33] shows that a significant amount of qubits are needed to add distinguishability in integer-to-bit mappings, such as the gray code encoding.

## 4 Conclusions

We have proposed an alternative method for the digital simulation of linear-optical networks by using the property that suitably entangled fermions can effectively behave like bosons. Unlike other existing B2QE protocols, our approach provides a simple and intuitive extension of an ideal bosonic system to a non-ideal one by introducing additional internal degrees of freedom. As a proof of concept, we designed quantum circuits for generating the Hong-Ou-Mandel dip by varying particle distinguishability. We successfully executed a digital simulation using the IBM Quantum and IonQ cloud services for the ideal boson case. For the partially distinguishable boson case, we showed the validity of our scheme with a classical simulation using Qiskit's qasm.

The obvious extension of our B2QE approach would be the non-number-conserving bosonic system simulations, such as Gaussian boson sampling [34] and molecular simulations [35]. However, confining the infinite bosonic Hilbert space to the finite qubit Hilbert space will intrinsically generate errors for the non-number-conserving bosonic problems. In future work, we will attempt to optimize the required resources and errors induced by the confinement. We

also intend to develop another efficient quantum algorithm for computing the matrix permanent [36] based on our B2QE protocol. With the help of the new B2QE protocol, we envisage developing efficient qubit-based quantum algorithms for bosonic systems, e.g., the boson sampling with nonideal photons, the Bose-Hubbard model, and the spin-boson model.

## Acknowledgments

**Funding information**   This work was partly supported by the National Research Foundation of Korea (NRF-2019R1I1A1A01059964, 2020M3H3A1110365, 2021M3H3A103657313, NRF-2022M3H3A106307411, NRF-2023M3K5A1094805, NRF-2023M3K5A1094813, RS-2023-00245747). This work was partly supported by Institute for Information & communications Technology Promotion (IITP) grant funded by the Korea government(MSIP) (No. 2019-0-00003, Research and Development of Core technologies for Programming, Running, Implementing and Validating of Fault-Tolerant Quantum Computing System) and Education and Training Support Program of the Quantum Information Science Research Support Center, funded by the Ministry of Science Research and ICT of the Korean government. We also acknowledge support from the Samsung Advanced Institute of Technology.

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
