# Peer review of "Quantum circuit simulation of linear optics using fermion to qubit encoding"

_SciPost Physics Core, doi:SciPost Phys. Core 7, 042 (2024)_

## Round 1 · Referee Report · Anonymous (Referee 2) · 2024-4-14

Report
The protocol's efficacy is demonstrated through the simulation of the Hong-Ou-Mandel dip on both IBM Quantum and IonQ cloud platforms. The results are convincingly presented, and the protocol, while generally decently articulated, can be complex and somewhat challenging to follow. Additional explanatory detail, especially concerning the transformations and derivations, would enhance comprehension for readers less familiar with the topic.
My primary concern regards the originality and impact of the manuscript. As other reviewers have pointed out, numerous boson-to-qubit mappings (Refs. [12-18]) capable of simulating the Hong-Ou-Mandel dip already exist. Although the authors highlight their method's ability to handle partially distinguishable bosons, they do not provide concrete examples in which their method outperforms existing ones. For this work to stand out sufficiently for publication in this journal, it is in my opinion necessary to present a rigorous demonstration of its advantages over current methods. In its current form, the manuscript, while valid, appears to offer only incremental advancements.
Requested changes
1- example of situation showcasing the advantages of the method, e.g where existing methods would fail while the proposed protocol does not.
Recommendation
Ask for minor revision
Author: Joonsuk Huh on 2024-05-18 [id 4492]
(in reply to Report 1 on 2024-04-14)
Reviewer 1. 1- example of situation showcasing the advantages of the method, e.g where existing methods would fail while the proposed protocol does not.
Answer: The reviewer's claim is correct that other B2QE protocols can simulate HOM dip in principle. However, our protocol can achieve that more efficiently than other protocols based on integer-to-bit encodings optimized for ideal bosons. Our protocol is organized to preserve the exchange symmetry of bosons, resulting in a straightforward and efficient simulation of distinguishable bosons. As far as we know, the only research that has simulated HOM effect with digital quantum computers is Ref. [33] (recently posted in arXiv), which, however, is limited to ideal boson simulations with the gray code and acknowledges that a more general HOM dip simulation would require a significant amount of additional qubits. We have added a remark at the end of section III to demonstrate that our method can efficiently simulate the scattering process of distinguishable bosons. We left a remark in our manuscript (highlighted in blue) as follows:
"It is worth mentioning that our example of the HOM dip simulation with distinguishable bosons shows the advantage of our scheme over other integer-to-bit mappings in, e.g., Ref. [13-19]. Comparing Eq. (27) with Eq. (36), we see that bosonic system with a 2-dimensional internal degree of freedom is directly simulated by adding one copy of 4 qubits. Since our mapping from the bosonic system to qubits is set to preserve the exchange symmetry, the generalization from ideal bosons to distinguishable bosons is straightforward. Moreover, we do not need Schur transformation gates as in Ref. [12], hence more efficient. On the other hand, recent research on the digital simulation of the HOM experiment with ideal photons in Ref. [33] shows that a significant amount of qubits are needed to add distinguishability in integer-to-bit mappings, such as the gray code encoding."
Attachment:
Digital_simulation_of_linear_optics__Copy_revised18052024.pdf
Anonymous on 2023-11-07 [id 4098]
Dear Editor:
Thank you for your invitation for resubmitting the revised version of the manuscript. We appreciate the reviewer's constructive comments on our manuscript. We have addressed all reviewer's questions and improved our manuscript for clarity. We hope this revised manuscript will be suitable for publication in the SciPost Physics Core. The changes are highlighted in bold in the main text.
Sincerely, Joonsuk Huh
Reviewer #1 : <GENERAL COMMENTS> 1) Reviewer’s comment: In the case of HOM experiment with no internal degrees of freedom, there is no discussion of the results, we just see the histogram plot of the probabilities. How does this relate with the expected results? Are the results satisfactory? Are there errors? Why? etc. Presumably, we need some notion of fidelity or similar to analyze it.
Author’s response: We agree that Fig. 5 needs more explanation, which is supposed to demonstrate the photon bunching effect. According to Eq. 33, only photon bunching states, i.e., |1100> and |0011>, are expected to have output probabilities of 50%. However, the actual quantum simulation results shown in Fig. 5 contain significant errors due to measurement errors and the imperfect single-qubit and two-qubit gates. We have added an explanation for Fig. 5 (p6, 2nd column, 3rd paragraph) to provide a more detailed description of the results; we have included the results of tomography and fidelity calculations in Supplementary Materials (SM). We provide tomography and fidelity results in SM rather than in the main text because our focus is not on the simulation itself but on the new methodology (boson to qubit map). As it has been a while since we prepared the previous simulation results, the IBMQ device we used has been retired. We used other available IBMQ devices for the tomography. We obtained results using four IBMQ devices (ibm_brisbane, ibm_perth, ibm_lagos, and ibm_nairobi) and one IONQ device. Thus, we also modified Fig. 5 with a new data set accordingly. Fig. 5 (a) shows a simulation result using ibm_brisbane, and we changed the caption accordingly.
2) Reviewer’s comment: Regarding the experimental setup, IBM Quantum possess several devices. Why they choose london in particular? How many qubits? What are the technical specifications? -gates and measurement errors, quantum volume, connectivity etc. The same goes with IONQ.
Author’s response: Although not explicitly stated in the paper, we chose ibmq_london because we ran our simulations multiple times on all IBM Quantum devices accessible then and presented the best results. We agree that we need to provide information on the device's technical specifications because the available quantum devices can vary. Since "ibmq_london" has been retired, we conducted a new simulation using the currently available devices. We summarized the technical specifications of all five quantum computers from IBMQ and IONQ in SM. The IBMQ device, ibm_brisbane, produced the histogram in the main text.
3) Reviewer’s comment: I believe this case has been already considered in the literature, see reference [17]. How does the results here are related to the ones in this reference?
Author’s response: Our approach is distinct from the work in Ref. [17] in two aspects. First, while Ref. [17] uses the boson-qubit mapping proposed in Somma et al. Proc. SPIE 5105, Quantum Information and Computation 96, (2003), we propose a new boson-qubit mapping via a F2QE protocol (JW transformation). Our mapping has an advantage in simulating bosons with internal degrees of freedom, as mentioned in the introduction (p1, 2nd column, 2nd paragraph). Second, while Ref. [17] simulating beam-splitter and squeezing operations of indistinguishable photons with IBMQ, we used our mapping protocol to simulate HOM dip in which bosons can have distinguishability. To manifest the differences, we added a slightly detailed explanation on Ref. [17] (p1, 2nd column, 1st paragraph).

---

## Round 1 · Author Response

Dear Editor:
Thank you for your invitation for resubmitting the revised version of the manuscript. We appreciate the reviewer's constructive comments on our manuscript. We have addressed all reviewer's questions and improved our manuscript for clarity. We hope this revised manuscript will be suitable for publication in the SciPost Physics Core. The changes are highlighted in bold in the main text.
Sincerely, Joonsuk Huh
Reviewer #1 : <GENERAL COMMENTS> 1) Reviewer’s comment: In the case of HOM experiment with no internal degrees of freedom, there is no discussion of the results, we just see the histogram plot of the probabilities. How does this relate with the expected results? Are the results satisfactory? Are there errors? Why? etc. Presumably, we need some notion of fidelity or similar to analyze it.
Author’s response: We agree that Fig. 5 needs more explanation, which is supposed to demonstrate the photon bunching effect. According to Eq. 33, only photon bunching states, i.e., |1100> and |0011>, are expected to have output probabilities of 50%. However, the actual quantum simulation results shown in Fig. 5 contain significant errors due to measurement errors and the imperfect single-qubit and two-qubit gates. We have added an explanation for Fig. 5 (p6, 2nd column, 3rd paragraph) to provide a more detailed description of the results; we have included the results of tomography and fidelity calculations in Supplementary Materials (SM). We provide tomography and fidelity results in SM rather than in the main text because our focus is not on the simulation itself but on the new methodology (boson to qubit map). As it has been a while since we prepared the previous simulation results, the IBMQ device we used has been retired. We used other available IBMQ devices for the tomography. We obtained results using four IBMQ devices (ibm_brisbane, ibm_perth, ibm_lagos, and ibm_nairobi) and one IONQ device. Thus, we also modified Fig. 5 with a new data set accordingly. Fig. 5 (a) shows a simulation result using ibm_brisbane, and we changed the caption accordingly.
2) Reviewer’s comment: Regarding the experimental setup, IBM Quantum possess several devices. Why they choose london in particular? How many qubits? What are the technical specifications? -gates and measurement errors, quantum volume, connectivity etc. The same goes with IONQ.
Author’s response: Although not explicitly stated in the paper, we chose ibmq_london because we ran our simulations multiple times on all IBM Quantum devices accessible then and presented the best results. We agree that we need to provide information on the device's technical specifications because the available quantum devices can vary. Since "ibmq_london" has been retired, we conducted a new simulation using the currently available devices. We summarized the technical specifications of all five quantum computers from IBMQ and IONQ in SM. The IBMQ device, ibm_brisbane, produced the histogram in the main text.
3) Reviewer’s comment: I believe this case has been already considered in the literature, see reference [17]. How does the results here are related to the ones in this reference?
Author’s response: Our approach is distinct from the work in Ref. [17] in two aspects. First, while Ref. [17] uses the boson-qubit mapping proposed in Somma et al. Proc. SPIE 5105, Quantum Information and Computation 96, (2003), we propose a new boson-qubit mapping via a F2QE protocol (JW transformation). Our mapping has an advantage in simulating bosons with internal degrees of freedom, as mentioned in the introduction (p1, 2nd column, 2nd paragraph). Second, while Ref. [17] simulating beam-splitter and squeezing operations of indistinguishable photons with IBMQ, we used our mapping protocol to simulate HOM dip in which bosons can have distinguishability. To manifest the differences, we added a slightly detailed explanation on Ref. [17] (p1, 2nd column, 1st paragraph).

---

## Round 2 · Author Response

Reviewer 1. 1- example of situation showcasing the advantages of the method, e.g where existing methods would fail while the proposed protocol does not.
Answer: The reviewer's claim is correct that other B2QE protocols can simulate HOM dip in principle. However, our protocol can achieve that more efficiently than other protocols based on integer-to-bit encodings optimized for ideal bosons. Our protocol is organized to preserve the exchange symmetry of bosons, resulting in a straightforward and efficient simulation of distinguishable bosons. As far as we know, the only research that has simulated HOM effect with digital quantum computers is Ref. [33] (recently posted in arXiv), which, however, is limited to ideal boson simulations with the gray code and acknowledges that a more general HOM dip simulation would require a significant amount of additional qubits. We have added a remark at the end of section III to demonstrate that our method can efficiently simulate the scattering process of distinguishable bosons. We left a remark in our manuscript (highlighted in blue) as follows:
"It is worth mentioning that our example of the HOM dip simulation with distinguishable bosons shows the advantage of our scheme over other integer-to-bit mappings in, e.g., Ref. [13-19]. Comparing Eq. (27) with Eq. (36), we see that bosonic system with a 2-dimensional internal degree of freedom is directly simulated by adding one copy of 4 qubits. Since our mapping from the bosonic system to qubits is set to preserve the exchange symmetry, the generalization from ideal bosons to distinguishable bosons is straightforward. Moreover, we do not need Schur transformation gates as in Ref. [12], hence more efficient. On the other hand, recent research on the digital simulation of the HOM experiment with ideal photons in Ref. [33] shows that a significant amount of qubits are needed to add distinguishability in integer-to-bit mappings, such as the gray code encoding."
Answer: The reviewer's claim is correct that other B2QE protocols can simulate HOM dip in principle. However, our protocol can achieve that more efficiently than other protocols based on integer-to-bit encodings optimized for ideal bosons. Our protocol is organized to preserve the exchange symmetry of bosons, resulting in a straightforward and efficient simulation of distinguishable bosons. As far as we know, the only research that has simulated HOM effect with digital quantum computers is Ref. [33] (recently posted in arXiv), which, however, is limited to ideal boson simulations with the gray code and acknowledges that a more general HOM dip simulation would require a significant amount of additional qubits. We have added a remark at the end of section III to demonstrate that our method can efficiently simulate the scattering process of distinguishable bosons. We left a remark in our manuscript (highlighted in blue) as follows:
"It is worth mentioning that our example of the HOM dip simulation with distinguishable bosons shows the advantage of our scheme over other integer-to-bit mappings in, e.g., Ref. [13-19]. Comparing Eq. (27) with Eq. (36), we see that bosonic system with a 2-dimensional internal degree of freedom is directly simulated by adding one copy of 4 qubits. Since our mapping from the bosonic system to qubits is set to preserve the exchange symmetry, the generalization from ideal bosons to distinguishable bosons is straightforward. Moreover, we do not need Schur transformation gates as in Ref. [12], hence more efficient. On the other hand, recent research on the digital simulation of the HOM experiment with ideal photons in Ref. [33] shows that a significant amount of qubits are needed to add distinguishability in integer-to-bit mappings, such as the gray code encoding."

---

## Editorial Decision

published